# Body Composition and Its Outcomes and Management in Multiple Sclerosis: Narrative Review

**DOI:** 10.3390/nu17061021

**Published:** 2025-03-14

**Authors:** Ariel Kidwell-Chandler, Justin Jackson, Brenda Jeng, Stephanie L. Silveira, Lara A. Pilutti, Paul R. Hibbing, Robert W. Motl

**Affiliations:** 1Department of Kinesiology and Nutrition, College of Applied Health Sciences, University of Illinois Chicago, Chicago, IL 60612, USA; akidwe2@uic.edu (A.K.-C.); jjacks69@uic.edu (J.J.); bjeng@uic.edu (B.J.); phibbing@uic.edu (P.R.H.); 2Department of Management, Policy, and Community Health, University of Texas Health Science Center at Houston, Houston, TX 77030, USA; stephanie.l.silveira@uth.tmc.edu; 3Interdisciplinary School of Health Sciences, Brain and Mind Research Institute, University of Ottawa, Ottawa, ON K1N 6N5, Canada; lpilutti@uottawa.ca

**Keywords:** multiple sclerosis, body composition, obesity, adipose tissue, bone mineral density, lean soft tissue mass, dual-energy X-ray absorptiometry

## Abstract

*Background*: There is emerging interest in obesity and its prevalence, outcomes, and management in people with multiple sclerosis (MS). Body mass index (BMI) is the traditional marker of obesity in MS, whereas body composition, inclusive of specific body tissue compartments (e.g., fat, bone, and muscle), is often overlooked despite its relevance. *Objective:* This narrative review (a) underscored the use and utility of dual-energy X-ray absorptiometry (DEXA) as an accurate and reliable measure of body composition; (b) thematically analyzed and synthesized the current evidence regarding body composition (using DEXA); and (c) determined gaps to be addressed in future research. *Methods*: The structure and reporting of this narrative review followed the guiding criteria outlined in the Scale for the Assessment of Narrative Review Articles (SANRA). The relevant literature for this narrative review was identified via a PubMed search utilizing combined search terms such as ‘body composition’ and ‘multiple sclerosis’. The identified research was then organized by the authors into major themes and sub-themes. The articles described within the narrative review were based on saturation of the identified themes and sub-themes. *Results*: Three major themes were identified, namely (1) comparison of body composition between people with MS and non-MS controls (2 meta-analyses); (2) examination of the relationships between body composition and a range of outcomes (14 cross-sectional studies); and (3) interventions that report and/or target body composition in MS (11 clinical trials). *Conclusions*: This narrative review mapped the existing evidence regarding body composition in MS, and posits body composition as a novel, informative, and targeted concept for this population. The narrative review underscores the importance of randomized controlled trials that focus on body composition as a significant and modifiable outcome. Such research could improve the understanding of obesity and poor body composition in MS and identify useful clinical recommendations for diagnosis and management.

## 1. Introduction

Obesity, defined by excessive adiposity, and commonly quantified as a body mass index (BMI; weight in kilograms/height in meters squared [kg/m^2^]) of ≥30 kg/m^2^, is a prevalent disease that can undermine health, well-being, and longevity [1,2]. The 2021–2023 cycle of the National Health and Nutrition Examination Survey (NHANES) reported the prevalence of obesity as 40.3% of adults in the general United States (US) population, and this is seemingly increasing compared with earlier estimates of 30.5% from the 1999–2000 NHANES [3,4]. Furthermore, obesity based on BMI is a health risk factor associated with mental and physical health conditions, including depression, type 2 diabetes, and cardiovascular disease, as well as several types of cancers, and all-cause mortality [5,6].

Some evidence suggests that the prevalence and burden of obesity, typically based on BMI, may be even more influential in people living with chronic diseases and conditions such as multiple sclerosis (MS) [7,8]. MS is a chronic, progressive, and often disabling immune-mediated disease of the central nervous system, and researchers have hypothesized a central role of obesity in MS [8,9,10,11,12]. Indeed, obesity and its related comorbidities may portend serious negative consequences for this disease, its course and manifestations, and mortality [8].

The current understanding of obesity in MS is largely derived from BMI predicated on its relationships with disease development and progression, symptoms, and comorbid conditions [8,12,13,14,15,16]. Nevertheless, the evidence overall is generally unclear regarding the definitive role of obesity in MS [8,10]. This lack of clarity may be associated with the application of anthropometric measures of body composition, such as BMI, as the primary metric for characterizing adiposity and diagnosing obesity [17]. The reliance on BMI is understandable, as it is a simple and accessible measure, yet BMI provides limited quantification of body composition as it is based solely on height and weight. BMI therefore does not reflect compartments or tissues of the body (e.g., fat, bone, muscle) [17,18,19,20]. BMI cut points further suffer from methodological and conceptual drawbacks because the cut-points generalize poorly across ethnic/racial groups, diseases, and disease states [21,22,23,24]. Indeed, people with MS are less physically active, more sedentary, and have higher rates of osteoporosis and lower bone mineral density than non-MS controls, and those variables influence weight-by-height outcomes and thus challenge the suitability and comparability of BMI as a surrogate for adiposity [25,26,27]. BMI, therefore, may not be as useful for research in MS, and it may similarly be insufficient for guiding clinical decisions with patients, particularly regarding behavioral approaches for lifestyle and disease management (e.g., physical activity and diet) [28,29,30].

There is a need to move the field beyond BMI as a commonly applied measure of adiposity in MS and instead focus on a more precise methodology for contextualizing adiposity and other critical body tissue compartments (i.e., bone and muscle). This narrative review presents an overview of body tissue compartments (hereafter referred to as body composition) and discusses the application, strength, and limitations of dual-energy X-ray absorptiometry (DEXA) as an accurate and reliable approach for measuring body composition [17,18,19,20,31]. This narrative review then offers a thematic synthesis of the existing evidence base and follows the criteria outlined in the Scale for the Assessment of Narrative Review Articles (SANRA) [32]. The objective of this narrative review is the provision of a foundational overview that will guide future research on obesity and its management in MS through an integrative perspective. This objective moves the field beyond BMI and presents body composition as a novel and significant approach for researchers and clinicians in MS.

## 2. Concepts and Measurement of Body Composition

Body composition is appropriately defined as the study of different components or compartments in the human body, and it represents the morphological domain of health-related fitness [33]. The selection and study of compartments vary by model and can be described based on two-, three-, four-, and multicompartmental models with each having distinct assumptions/algorithms [20]. The two-compartmental model (2C) partitions body mass into fat mass and fat-free mass, based on the discrete densities of fat mass, fat-free mass, and water [20]. Common 2C methods of measurement are hydrodensitometry, air displacement plethysmography, and hydrometry [20]. The three-compartment model (3C) incorporates an additional component by subdividing fat-free mass into bone mineral and lean soft tissue mass [20]. DEXA via the transmission of low-energy X-rays through the body is a common approach for the 3C model [20]. Both the four-compartmental model (4C) and the multicompartmental model provide even greater specificity of body composition [20]. The 4C model includes fat, mineral, protein, and total body water, and the multicompartmental model details major elemental composition (e.g., calcium, sodium, potassium, chloride, nitrogen, etc.) [20]. The 4C and multicompartmental models are infrequently applied in research and clinical settings based on constraints of time, cost, training, and equipment [20].

The caliber of body composition measurement is dependent on the application of systematic and rigorous methodology utilizing accurate and reliable tools that are reasonably practicable [17,18,19,20]. The measurement should provide insights regarding disease risk, state, and/or status. DEXA is typically considered a gold-standard measure for body composition [17,18,19,20]. DEXA is regarded as user-friendly, as the administration is quick (i.e., 10–20 min) and generally straightforward [17]. DEXA presents minimal risk, as there is relatively minimal radiation exposure (<5 mrem or 0.05 mSv) [17]. DEXA has the capability of accurately and reliably measuring adiposity (i.e., fat mass), bone mineral density (i.e., surrogate for bone mass), and lean soft tissue mass (i.e., surrogate for muscle mass), and can provide detailed regional body composition [17,18,19,20]. Indeed, DEXA distinguishes the quantities and proportions of tissue types that are commonly implicated in health-related fitness [33]. There are some important considerations when using DEXA, including (a) variable estimations by model/machine, (b) some participants may not fit on the table/within the scan area (although there are strategies to mitigate this), and (c) the machine may not be suitable for all studies depending on sample size, accessibility, and budget [17,18,19,20]. Below, we describe the tissue types that DEXA measures and the importance of these tissue types within the human body, as well as the respective advantages and disadvantages of DEXA for capturing the tissue types.

### 2.1. Adiposity (Fat Mass)

Adipose tissue is a specialized type of connective tissue composed of fat cells called adipocytes [34]. Broadly, adipose tissue functions as cushioning, insulation, and energy storage [34]. Adipose tissue is considered an endocrine and secretory organ, as it participates in specific autocrine, paracrine, and endocrine actions that are central to core processes including inflammatory response and metabolic function [34,35,36,37]. Adipose tissue has a high degree of plasticity such that it can expand or shrink with energy intake (i.e., calories in) and demand (i.e., calories out), and excessive adiposity (i.e., accretion of hypertrophic adipocytes) increases the risk of a hypoxic environment, insulin resistance, and extracellular matrix accumulation [37]. Furthermore, the proportion and location of adipose tissue on the body are relevant for health and functioning. For example, visceral adipose tissue (VAT), characterized by adipose tissue surrounding intra-abdominal organs, is generally associated with morbidity and mortality risks (e.g., metabolic disturbances, several cancers, and cardiovascular disease) compared with subcutaneous adipose tissue (SAT) (i.e., adipose tissue between skin and muscles) [38].

DEXA does not provide specific details regarding the number of hypertrophic cells or precise cellular actions, but it does provide valuable information for quantifying the relative mass of adipose tissue and its location on the body (e.g., whole body, left and right arms, left and right legs, trunk, android, gynoid) [38,39]. This allows for comparative insights within participants and between subgroups of participants (e.g., MS versus non-MS controls) regarding adiposity. Nevertheless, there are some limitations of DEXA for quantifying adiposity. One consideration is that some participants may be too large for the table, thus, researchers may need to utilize scanning techniques whereby partial/offset scans are acquired to generate complete whole-body scans [40]. This necessitates that researchers reference the manufacturer’s guidelines and cite the hardware, software, and general methodology utilized in any reporting. DEXA further can accurately detail quantities and proportions of fat mass, yet certain measures, including VAT, are indirectly measured, and researchers should interpret and report the results accordingly [38].

### 2.2. Bone Mineral Density (Surrogate for Bone Mass)

Bone, a mineralized connective tissue, provides structure, support, and protection for the human body, and further plays a role in blood cell and hormonal production as well as a storage site for key minerals and vitamins (e.g., calcium, vitamin D) [41]. Bone is classified as an endocrine organ, as it is implicated in the regulation of both local bone metabolism and overall metabolic functions that are important for global energy homeostasis [41,42]. Bone can be differentiated by organic matter (i.e., type I collagen and non-collagenous proteins), as well as inorganic matter (i.e., bone mineral) [41]. Bone mass, often quantified via bone mineral density (BMD), a measure of the relative density of calcium and other bone minerals, provides insight into the strength of bones [43]. BMD is valuable for predicting fracture risk (i.e., lower BMD equates to higher fracture risk) as well as for diagnosis, severity classification, and monitoring of conditions such as osteoporosis [43].

DEXA can accurately report whole-body BMD, from which regional (e.g., hip, femoral, lumbar spine, and wrist) BMD can be derived accordingly [43,44]. DEXA can further report the absolute quantity of minerals in a bone region, called bone mineral content (BMC) [44]. Measures of bone from DEXA provide specificity for targeted treatment (e.g., resistance training), as well as comparative insight within participants and between subgroups relative to bone health and/or other potentially relevant disease-related outcomes [43,44]. DEXA is considered a gold-standard method for obtaining BMD, yet there are user considerations [43,44]. Participants may be too tall for the scan area, and similar precautions to those with adiposity should be taken whereby partial/offset scans can be utilized for generating a whole-body scan [40,45]. Researchers are advised to follow manufacturer guidelines regarding positioning, as incorrect positioning may adversely impact BMD calculations. Artifacts including kidney stones, dense metals (e.g., prosthetic joints), and prior administration of radionuclides, for example, can result in error. As such, screening participants is recommended [42,46].

### 2.3. Lean Soft Tissue (Surrogate for Muscle Mass)

Skeletal muscle is the heaviest organ in the body, comprising roughly 50% of total body weight and 50–75% of all body proteins [47,48]. Skeletal muscles are attached to bone and are responsible for movement, heat production, and posture [47,48]. Skeletal muscles are also collectively considered an endocrine organ, based on the secretion of cytokines and other proteins called myokines that are pointedly involved in inflammatory processes [49,50,51]. Indeed, the imbalance amongst these substances towards a pro-inflammatory state yields a favorable environment for numerous comorbidities such as sarcopenia, depression, type 2 diabetes, cardiovascular disease, several cancers, as well as dementia [49,50,51]. Muscle mass is dependent on the balance between protein synthesis and degradation, and the quantity of skeletal muscle, particularly in relation to the proportions of other tissues (i.e., fat and bone), may provide insight into an individual’s behaviors (e.g., physical activity and diet), and information about disease state or status [52].

DEXA does not directly assess skeletal muscle mass but rather reports the surrogate ‘lean soft tissue mass’ [53]. Lean soft tissue mass via DEXA includes skeletal muscle, viscera, and fluids, and, like fat and bone, can be reported regionally or for the whole body [53]. Reporting lean soft tissue mass rather than skeletal muscle mass is a notable limitation; however, lean soft tissue mass may be a helpful reference measure for muscle mass, particularly when combined with other functional and strength-related assessments (e.g., grip strength) [31,53,54]. Researchers should be aware of these limitations when interpreting and reporting results and may consider adding additional functional and strength-related measures that capture health-related fitness.

## 3. Methods

### 3.1. Quality Assessment of Narrative Reviews

The quality of narrative reviews can be evaluated via the SANRA assessment and six quality criteria metrics: (1) justification of the article’s importance for the readership, (2) statement of concrete aims or formulation of questions, (3) description of the literature search, (4) referencing, (5) scientific reasoning, and (6) appropriate presentation of data [32]. The first two criteria were satisfied in the Introduction and below (i.e., *Rationale and Objective*), the third and fourth criteria are further addressed below (i.e., *Literature Search and Study Selection*), and key statements throughout this review are referenced (i.e., fourth criteria), and evidence and data are presented in the Results with primary findings summarized and interpreted in the Discussion (i.e., fifth and sixth criteria) [32].

### 3.2. Rationale and Objective

We undertook a narrative review, specifically an empirical integrative review, that represents a thematic synthesis of the current evidence regarding body composition in MS, with a focus on the use of DEXA [55]. Importantly, we note that this narrative review did not intend to comprehensively synthesize all research on this topic, but rather present an argument for the use of accurate and reliable measures of body composition, such as DEXA, and provide a broad overview of the state of the evidence necessary for the advancement of this research in MS. The objective is the provision of a foundational overview that will guide future research on obesity and its management in MS through an integrative perspective. This objective presents body composition as a novel and significant approach for researchers and clinicians in MS. As noted above, this description of the objective and importance, in combination with the description provided in the Introduction, fulfills the first two SANRA criteria [32].

### 3.3. Literature Search and Study Selection

This narrative review identified relevant literature via a PubMed search utilizing combined key terms such as ‘body composition’ and ‘multiple sclerosis’; this is consistent with the third SANRA criterion [32]. The search was inclusive of the time period of inception of PubMed through 1 December 2024. The authors then reviewed the identified literature and agreed on major themes and sub-themes using a modified Delphi approach for synthesis [56]. There were no predetermined inclusion or exclusion criteria. Literature was selected per theme and sub-theme independently by one section lead (i.e., one lead per theme identified). The collection of selected literature was then independently re-evaluated by a second individual to ensure adequate saturation of the focal topic, and a third individual addressed discrepancies. This was achieved by prioritizing peak evidence (i.e., meta-analyses), and including both cross-sectional studies and clinical trials that reported DEXA outcomes across multiple body tissue compartments. No articles were screened out based on publication date, but priority was given to research published within the past 15 years so that the most up-to-date evidence was included. This fulfilled the fourth, fifth, and sixth SANRA criteria [32].

## 4. Results

Three major themes were identified by the research team, namely (1) studies that compared body composition between people with MS and non-MS controls, (2) studies that examined the relationships between body composition and disease- and health-related outcomes, and (3) interventions that report body composition outcomes/discuss management of body composition in MS. Below, we review each major theme, along with their corresponding sub-themes.

### 4.1. Body Composition in Multiple Sclerosis Versus Controls

There is considerable evidence regarding BMI in MS compared with the general population (i.e., non-MS controls). Indeed, one systematic review and meta-analysis examined differences in BMI between people with MS and non-MS controls [57]. The systematic literature review yielded 25 studies that compared BMI outcomes between people with MS (*n* = 2914) and non-MS controls (*n* = 3314) [57]. The standardized mean difference (SMD) of BMI in people with MS (SDM = –0.25) was significantly lower than non-MS controls (SDM = –0.27) [57]. The researchers reported variation amongst the studies with respect to BMI and recommended the use of additional indicators of global mass in future research [57]. Results from this study are intriguing in that they indicate that BMI is lower in MS than in non-MS controls; however, it is unclear if this reflects differences in adiposity, bone, or lean soft tissues. This therefore provides a rationale for examining body composition between people with MS and non-MS controls using DEXA for a more accurate understanding of differences in tissue types.

Another recent systematic review and meta-analysis quantified compartment-specific differences between people with MS and non-MS controls as Glass’s delta (Δ) [58]. The systematic literature review yielded 37 studies that compared body composition outcomes between people with MS (*n* = 2127) and non-MS controls (*n* = 2330), and the studies typically measured body composition using DEXA (*n* = 30) [58]. Results indicated that people with MS had worse overall body composition than non-MS controls (∆ = −0.39), and the differences were particularly salient among those with higher disability status (i.e., significantly larger differences with higher disability levels) [58]. There were notable differences between people with MS and non-MS controls across body compartments [58]. People with MS had significantly more body fat than controls (∆ = −0.32) [58]. There were further differences in BMD, whereby people with MS had lower BMD than controls (∆ = −0.44) [58]. People with MS additionally had less lean soft tissue than controls (∆ = −0.38) [58]. Such results provide an overall picture of body composition based on a 3C model using DEXA, and the apparent discrepancy between the two meta-analyses further underscores the importance of examining body composition in MS [57,58].

### 4.2. Body Composition via DEXA and Disease- and Health-Related Outcomes in Multiple Sclerosis

Based on the aforementioned differences in body composition between MS and non-MS controls, this section reviews the association between body composition outcomes from DEXA and biomarkers, cognition, mobility, symptoms, fitness, and quality of life in MS. Figure 1 and Appendix A provide an overview and summary of associations by tissue type (adiposity [i.e., fat mass], BMD [i.e., surrogate for bone mass], and lean soft tissue mass [i.e., surrogate for muscle mass]) with outcomes of MS.

#### 4.2.1. Biomarkers

Blood-based biomarkers have commonly been measured for insights about disease state and/or status, and some researchers have examined associations with DEXA-based body composition in MS. One study examined the relationship between measures of adiposity from DEXA and C-reactive protein (CRP), a global marker of systemic inflammation, in MS and non-MS controls using non-parametric correlations expressed as Spearman’s rho (ρ) [59]. Among those with MS, whole-body fat mass (ρ = 0.39) and percent (%) body fat (ρ = 0.44) were significantly associated with CRP; however, trunk fat mass and % trunk fat were not significantly associated with CRP [59]. By comparison, all adiposity measures including whole-body fat mass (ρ = 0.65), % body fat (ρ = 0.58), trunk fat mass (ρ = 0.69), and % trunk fat (ρ = 0.64) were significantly associated with CRP in the non-MS control sample [59]. Results for both groups were adjusted for age, sex, and physical activity levels [59].

Another biomarker of interest is vitamin D, a lipid-soluble vitamin that is markedly lower in MS and associated with MS risk and disease activity [60]. We located three studies that examined the relationship between levels of BMD of the femoral neck and lumbar spine and vitamin D levels, and all three reported no statistically significant relationship [61,62,63].

#### 4.2.2. Cognition

There is consistent evidence for cognitive impairment in MS, yet there is limited information regarding behavioral approaches for its management [8,64,65,66]. There are data indicating that body composition is associated with cognitive impairment from other populations, and this has supported the study of potential associations in MS [67,68,69,70,71,72].

We located three studies examining body composition and cognitive outcomes in MS [67,68,69]. Two of the studies examined the relationship between adiposity from DEXA and cognitive outcomes [67,69]. After adjusting for variables such as age, sex, disability status, and/or disease duration, no significant associations were noted between whole-body fat mass or % body fat with outcomes from a battery of cognitive assessments that measured visuospatial learning and memory (Brief Visuospatial Memory Test—Revised [BVMT-R]), verbal learning and memory (California Verbal Learning Test—Second Edition [CVLT-2]), or information processing speed (Symbol-Digit Modalities Test [SDMT]) [67,69].

All three studies examined BMD and cognitive outcomes [67,68,69]. Two studies reported no significant associations between whole-body BMD and BMC and the cognitive assessments (BVMT-R, CVLT-2, and SDMT) after adjusting for variables such as age, sex, disability status, and/or disease duration [67,69]. The third study compared outcomes between people with cognitive impairment and those without cognitive impairment and reported that the cognitively impaired MS group had more cases of osteopenia or osteoporosis (59.3% versus 24.1%) and had significantly lower femur BMD (Cohen’s *d* = −0.68) but not lumbar spine BMD than the MS group without cognitive impairment [68]. In both groups combined, and after adjusting for disability status, femur BMD was significantly associated with visuospatial learning and memory assessments including BVMT-R Delayed Recall (*r* = 0.39), BVMT-R Total Learning (*r* = 0.41), and Judgement of Line Orientation Test (JLO) (*r* = 0.34) [68]. By comparison, there were no significant associations between BMD and measures of verbal learning and memory (Controlled Oral Word Association Test [COWAT]), executive functioning (Delis–Kaplan Executive Function System Scoring Test [DKFS]), or information processing speed (Paced Auditory Serial Addition Test—3s [PASAT]) [68].

Two of the studies examined the relationship between whole-body lean soft tissue mass and cognitive outcomes [67,69]. One of the studies reported no associations between whole-body lean soft tissue mass and visuospatial learning and memory (BVMT-R), verbal learning and memory (CVLT-2), or information processing speed (SDMT) after adjusting for age and disability status [67]. Bivariate associations were, however, noted between lean soft tissue mass and information processing speed (SDMT) in the second study, after adjusting for age, sex, disability status, and disease duration (ρ = 0.39) [69].

#### 4.2.3. Mobility

Mobility disability or reduced walking performance, characterized by slowed walking speed and decreased walking endurance, is common in MS [73]. Mobility disability can further be characterized by reduced movement efficiency and increased energy cost of walking (i.e., walking efficiency) in MS [74]. Within other populations, adipose tissue proportions (including SAT, VAT, and intra-muscular adipose tissue depots), muscular atrophy, and inadequate maintenance of BMD have been associated with mobility decline [75,76,77].

We located five studies that examined the relationship between body composition and mobility outcomes in MS that utilized measures including self-reported walking ability (12-Item Multiple Sclerosis Walking Scale [MSWS-12]), general mobility and/or balance (Star Excursion Balance Test [SEBT] and/or Timed Up and Go [TUG]), walking efficiency (energy cost of walking via measure of oxygen use while walking), walking speed (Timed 25-Food Walk [T25FW]), and/or walking endurance (Six-Minute Walk [6 MW]) [69,78,79,80,81]. All five studies investigated associations between adiposity and mobility outcomes [69,78,79,80,81]. Two studies reported no significant associations between whole-body fat mass or % body fat and mobility outcomes including MSWS-12, TUG, T25FW, and 6 MW, after adjusting for variables such as age, sex, disability status, and/or disease duration [69,81]. One study reported no significant associations between % body fat and either SEBT or 6 MW, however, that study did report significant associations between % body fat and TUG (*r* = 0.51) [79]. Significant associations were reported in another study, whereby after adjusting for age and disease duration, % body fat was positively associated with TUG (ρ = 0.35) and T25FW (ρ = 0.45) but was negatively associated with 6 MW (ρ = −0.44) [78]. The fifth study reported that % body fat was significantly and negatively associated with walking efficiency after adjusting for age, sex, disability status, step length, and cadence (ρ = −0.37) [80].

Three studies examined BMD and BMC with mobility outcomes [69,80,81]. In two of the studies, neither BMD nor BMC was significantly associated with the MSWS-12, TUG, T25FW, and 6 MW after adjusting for variables such as age, sex, disability status, and/or disease duration [69,81]. In the third study, whole-body BMD was significantly and negatively associated with walking efficiency after adjusting for age, sex, disability status, step length, and cadence (ρ = −0.37), although whole-body BMC was not significantly associated with walking efficiency [80].

Four studies investigated the relationship between lean soft tissue mass and mobility outcomes [69,79,80,81]. Three studies reported nonsignificant associations between whole-body lean soft tissue mass and mobility outcomes (MSWS-12, TUG, T25FW, 6 MW, and walking efficiency) after adjusting for variables such as age, sex, disability status, disease duration, step length and/or cadence [69,80,81]. One study, however, did report significant associations between leg lean soft tissue mass (*r* = 0.45) and leg lean soft tissue mass to body mass ratio (*r* = 0.53) with 6 MW performance, and a significant and negative association between leg lean soft tissue mass to body mass ratio and TUG (*r* = −0.53) [79]. In contrast, leg lean soft tissue mass was neither associated with TUG nor SEBT, and leg lean soft tissue mass to body mass ratio was not associated with SEBT [79].

#### 4.2.4. Symptoms

MS is associated with burdensome symptoms including anxiety, depression, fatigue, and pain [8,9]. Body composition outcomes might be associated with worse symptoms in MS based on research involving other populations [82,83,84,85].

We located three studies that examined body composition outcomes and the relationship with symptoms of anxiety, depression, fatigue, and pain in MS [69,86,87]. All three studies examined adiposity and symptom outcomes [69,86,87]. One study reported that whole-body fat mass was significantly associated with pain (McGill Pain Questionnaire [MPQ]) (ρ = 0.32), but not with anxiety (Hospital Anxiety and Depression Scale—Anxiety [HADS-A], depression (Hospital Anxiety and Depression Scale—Depression [HADS-D]), or fatigue (Modified Fatigue Impact Scale [MFIS]) after adjusting for age, sex, disability status, and disease duration [69]. Moreover, % body fat was not significantly associated with any of the symptoms including anxiety (HADS-A), depression (HADS-D), fatigue (MFIS), or pain (MPQ) in that study [69]. Another study included people with MS with and without depression and reported that the group with depression had significantly worse body fat % (OR = 1.07); however, there were no significant differences between the groups regarding whole-body fat mass and visceral fat level (VFL [a standardized measure of VAT]) [86]. The final study reported no significant bivariate associations between whole-body or site-specific (i.e., trunk, left and right arms, and left and right leg) % body fat and anxiety (HADS-A), depression (HADS-D), fatigue (MFIS and the Fatigue Severity Scale [FSS]), and pain (MPQ) after adjusting for age, sex, and disability status [87].

There were two studies that examined bivariate associations between BMD and symptom outcomes [69,87]. One study reported that neither whole-body BMD nor BMC was associated with anxiety (HADS-A), depression (HADS-D), fatigue (MFIS), or pain (MPQ) after adjusting for age, sex, disability status, and disease duration [69]. The other study further reported no significant bivariate associations between whole-body or site-specific (i.e., trunk, left and right arms, and left leg) BMD or BMD and anxiety (HADS-A), depression (HADS-D), fatigue (MFIS and the Fatigue Severity Scale [FSS]), and pain (MPQ) after adjusting for age, sex, and disability status [87].

There were two studies that examined bivariate associations between whole-body lean soft tissue mass and whole-body or site-specific (i.e., trunk, left and right arms, and left leg) % lean soft tissue mass and symptoms’ outcomes [69,87]. There were no significant associations between whole-body lean soft tissue mass and whole-body or site-specific % lean soft tissue mass and anxiety (HADS-A), depression (HADS-D), fatigue (MFIS and FSS), or pain (MPQ) after adjusting for variables such as age, sex, disability status, and/or disease duration [69,87]. Additionally, there were no significant differences between those with depression versus no depression and whole-body lean soft tissue mass or predictive muscle mass (PMM) [86].

#### 4.2.5. Fitness

There is consistent evidence that cardiorespiratory fitness and muscle strength outcomes are compromised in MS compared with non-MS samples, and this might be related to body composition based on associations with markers of cardiorespiratory fitness and muscular strength in the general population of adults [88,89,90,91].

One study examined the association between body composition and health-related fitness outcomes (cardiorespiratory and muscular) in MS [69]. Whole-body fat mass (ρ = −0.52) and % body fat (ρ = −0.56) were both significantly and negatively associated with cardiorespiratory fitness (peak oxygen uptake [VO_2peak_]) [69]. By comparison, neither whole-body fat mass nor % body fat was associated with muscular strength outcomes including grip strength, leg extensor peak torque (LE_peak_), and leg flexor peak torque (LF_peak_) [69]. The study further reported that whole-body BMD was associated with LE_peak_ (ρ = 0.35), but there were no significant associations between BMD and grip strength, VO_2peak_, or LF_peak_ [69]. Whole-body BMC, however, was related to grip strength (ρ = 0.53), LE_peak_ (ρ = 0.69), and LF_peak_ (ρ = 0.50), but like BMD, it was not associated with cardiorespiratory fitness (VO_2peak_) [69]. Whole-body lean soft tissue mass was significantly associated with all the strength outcomes including grip strength (ρ = 0.38), LE_peak_ (ρ = 0.48), and LF_peak_ (ρ = 0.38), but it was not significantly associated with cardiorespiratory fitness (VO_2peak_) [69]. Results in that study were adjusted for age, sex, disability status, and disease duration [69].

#### 4.2.6. Quality of Life

There are data suggesting that people with MS have lower health-related quality of life (HRQOL) than the general population and people living with other chronic diseases [92,93]. Studies in other populations have indicated that body composition may be associated with overall quality of life (QOL) and HRQOL [94,95,96].

We located two studies that examined the relationship between body composition and HRQOL [69,97]. One study examined the relationship between adiposity and HRQOL [69]. The study reported no significant associations between whole-body fat mass and either the physical domain of HRQOL (29-Item Multiple Sclerosis Impact Scale-Physical [MSIS-29—Physical]) or the psychological domain of HRQOL (29-Item Multiple Sclerosis Impact Scale-Psychological [MSIS-29—Psychological]) after adjusting for age, sex, disability status, and disease duration [69]. There were significant associations between % body fat and the psychological domain of HRQOL (MSIS-29—Psychological) (ρ = 0.34), but not the physical domain (MSIS-29—Physical) [69]. Both studies examined BMD and HRQOL [69,97]. Neither whole-body BMD nor whole-body BMC was significantly related to either domain of HRQOL (MSIS-29—Physical or MSIS-29—Psychological) after adjusting for age, sex, disability status, and disease duration in one study [69]. Femur BMD was significantly correlated with both domains of HRQOL (MSIS-29—Physical and MSIS-29—Psychological) in the other study [97]. Lumbar spine BMD was only significantly correlated with the physical domain of HRQOL (MSIS-29—Physical), but not the psychological domain (MSIS-29—Psychological) [97].

### 4.3. Interventions and Management of Body Composition in Multiple Sclerosis

Health behaviors such as physical activity and diet influence the disease course, expression, and management of MS, and there is evidence that health behaviors are associated with metrics of body composition in other populations [52,98,99,100,101]. This has, in part, supported the examination of behavioral interventions focusing on health behaviors for changing body composition in MS. This section reviews the effects of physical activity and diet-based interventions on body composition outcomes from DEXA. Appendix A provides an overview of the interventions, and the body composition results by tissue type (adiposity [i.e., fat mass], bone mineral density [i.e., surrogate for bone mass], and lean soft tissue mass [i.e., surrogate for muscle mass]).

#### 4.3.1. Physical Activity Interventions

Physical activity is defined as any bodily movement that results from skeletal muscle contraction and is associated with a substantial increase in energy expenditure, and this includes exercise training and free-living, lifestyle physical activity [102]. Exercise is safe for people with MS and represents a valuable rehabilitation strategy for managing symptoms and disease progression, yet there is limited evidence of the effects of exercise training on body composition in MS [103,104]. We identified four studies that administered aerobic exercise training interventions and reported body composition outcomes from DEXA [105,106,107,108]. One study examined the effects of 6 months of home-based, periodized, high-intensity interval exercise training (HIIT) on body composition outcomes in people with MS who had mild disability (EDSS < 4) and non-MS controls [105]. After the six-month intervention, there were no significant pre–post differences in whole-body fat mass, % body fat, or whole-body lean soft tissue mass in the MS or non-MS groups [105]. Another study compared the effects of a 12-week periodized HIIT program with a 12-week low-to-moderate intensity, continuous exercise training program for changes in body composition in people with MS who had mild disability (mean EDSS 2.3 ± 1.3) [106]. There were significant pre–post differences in whole-body fat mass (Δ = −3%, *p* = 0.002) and % body fat (Δ = −2%, *p* = 0.005) for the low-to-moderate intensity, continuous exercise training program, but not the periodized HIIT intervention [106]. There were no pre–post differences observed in whole-body lean soft tissue mass within or between groups [106].

Another study randomly assigned those with MS who had mild disability (EDSS range 1–5) into one of three, 12-week conditions: (1) high-intensity continuous cardiovascular cycle ergometer training plus resistance training (HCTR), (2) high-intensity interval cycle ergometer training group plus resistance training (HITR), or (3) a sedentary group (SED; i.e., no change to routine) [107]. There were significant pre–post differences in % body fat for both the HCTR (Δ = −2.5%, *p* = 0.02), and the HITR (Δ = −3.9%, *p* = 0.04) conditions, but not in the SED group [107]. Additionally, there were significant pre–post differences in whole-body lean soft tissue mass (Δ = 1.4%, *p* = 0.01) in the HITR group, but there were no changes in the HCTR condition nor the SED group [107]. The fourth study investigated the effects of a 10-month home-based structured running program (i.e., moderate-to-high intensity) on body composition outcomes between a sample of people with MS who had mild disability (EDSS < 5) and non-MS controls [108]. There were no significant pre–post differences in % body fat nor lean soft tissue mass in either group [108].

Lifestyle physical activity is another safe and beneficial intervention for people with MS, although there is little evidence regarding its impact on body composition [109]. One study examined body composition outcomes following a 6-month theory-based internet-delivered behavioral intervention promoting lifestyle physical activity (i.e., walking) in a sample of ambulatory people with MS versus a waitlist control condition [110]. There were no significant between-group post-intervention differences reported for either whole-body fat mass or % body fat [110]. Whole-body BMD and whole-body BMC post-intervention were significantly different between the groups favoring the intervention group when utilizing the unadjusted critical value (*p* < 0.05), but not the adjusted critical value (*p* < 0.008) [110]. Upon examining whole-body lean soft tissue mass, there were no significant post-intervention differences between groups [110].

Other approaches for managing symptoms and disease progression in MS involve rehabilitation strategies, such as Pilates, as a safe and effective mode for improving balance and strength, yet there are limited data available regarding body composition outcomes [111]. We located one study that reported body composition outcomes after the administration of either (1) Pilates twice per week plus massage therapy one time per week for 12 weeks or (2) massage therapy one time per week in a sample of people with MS [112]. There were no significant pre–post differences reported in whole-body fat mass, % body fat, or lean soft tissue mass between groups [112].

#### 4.3.2. Diet Interventions

Diet is considered an influential determinant of MS pathogenesis and progression, and there is evidence that diet-based interventions may provide promising benefits for symptoms and disease progression [99]. Overall, these interventions target nutrition status through the provision of advice, education, or delivery of the food component associated with a specific diet or tailored meal plan [113]. We located and reviewed four studies that reported body composition outcomes from DEXA based on intermittent fasting (IF) interventions or a theory-based behavioral intervention targeting the consumption of a low-glycemic index diet [114,115,116,117].

IF, characterized by a period of eating followed by a period of not eating or consuming energy (i.e., fasting), has gained considerable attention amongst clinical populations for weight management and potential anti-inflammatory effects; however, there are limited studies of IF in MS, and even fewer that report body composition outcomes using DEXA [99,118,119,120,121]. One study randomly assigned people with relapsing–remitting MS (RRMS) into one of three interventions: (1) 8-week controlled feeding IF-5:2 diet (i.e., 100% daily caloric needs five days/week and 25% daily caloric needs two consecutive days/week; IF-5:2); (2) 8-week controlled feeding calorie restriction diet (i.e., 78% daily caloric needs seven days/week; CR-78%); or (3) 8-week control diet with no calorie restriction (i.e., 100% daily caloric needs seven days/week; NR-100%) [114]. There were significant changes in whole-body fat mass between the IF-5:2 diet (Δ = −119.8 g/week) and the calorie restriction diet (CR-78%) (Δ = −330.1 g/week) (*p* = 0.03), but not when comparing the other conditions [114]. There were no significant average changes in % body fat, VAT mass, %VAT, or lean soft tissue mass when comparing any of the three conditions [114].

Another study tested the effects of an IF-5:2 diet on body composition outcomes using DEXA [115]. The sample of people with RRMS was randomly assigned into a 12-week period of (1) IF-5:2 diet (i.e., 500 calorie limit two non-consecutive days/week and usual intake with instructions to monitor intake/not overeat for five days) or (2) no restriction/no change to routine [115]. Pre–post whole-body fat mass was significantly different in the IF-5:2 (iCR) group (Δ = −2.21 g, *p* < 0.0001), but not within the no restriction/no change (NR) group or between groups [115]. There were significant pre–post differences in trunk fat mass in the iCR group (Δ = −1.4 g, *p* < 0.0001), but again, not within the NR group or between groups [115]. There were no significant pre–post differences reported in whole-body lean soft tissue mass within or between groups [115].

Few studies have examined the effects of IF using time-restricted eating in MS. One study examined the feasibility and acceptability of an 8-week period of an 8 h time-restricted eating (IF-8 h) intervention [116]. There were no significant pre–post differences reported for either % body fat or % lean soft tissue mass [116].

Dietary interventions delivered by nutrition professionals typically utilize the nutritional care process framework and applicably create tailored plans for individuals using evidence-based theory-informed interventions [113,122,123]. We located one study that utilized a theory-based behavioral intervention in MS for improving diet quality by administering a low glycemic index diet and subsequently reported body composition outcomes using DEXA [117]. There were significant differences in both pre–post whole-body fat mass (Δ = −1.94 kg, *p* = 0.006) and pre–post whole-body lean soft tissue mass (Δ = −0.72 kg, *p* = 0.035) following the 12-week telehealth intervention in the sample of RRMS participants [117].

#### 4.3.3. Combined Physical Activity and Diet-Based Intervention

One comprehensive approach for managing body composition in MS involves the provision of a combined physical activity and diet-based intervention, yet the existing behavioral interventions in MS have primarily focused on the independent effects of physical activity and nutrition. We located one study that combined both components and reported body composition outcomes using DEXA [124]. The study examined the effects of a 6-month theory-based group telehealth weight loss intervention in a sample of primarily RRMS participants [124]. Participants were randomly assigned into either a (1) home-based program to gradually increase physical activity to achieve a total of at least 150 min of moderate intensity physical activity five days/week plus a reduced calorie diet (i.e., 1200–1500 calories/day with composition guidelines) or (2) waitlist treatment as usual (i.e., were sent content regarding healthy eating and obesity prevention) [124]. Pre–post % body fat was significantly different in the intervention group (Δ = −3.4%) compared to the treatment as usual (TAU) group (Δ = −0.4%), (*p* = 0.001) [124].

## 5. Limitations

This narrative review is not without limitations. First, this narrative review offers a broad synthesis and mapping of the identified themes and sub-themes based on current evidence on body composition in MS. Accordingly, our research did not adhere to a particular set of guidelines necessary for systematic or scoping reviews. It may, therefore, be possible that the research team missed core themes and/or sub-themes that could be pertinent to body composition in MS or missed studies that may have been relevant and/or more conclusive. Second, the research team included interventions that were limited to physical activity and diet, and did not include pharmaceutical or surgical interventions, and future reviews may report the impact of such interventions on body composition in MS. Lastly, this narrative review primarily focused on body composition based on DEXA, and other measures of body composition may be more accessible or provide additional insight.

## 6. Future Research Directions

Body composition based on DEXA is a valuable outcome measure in MS, particularly when compared with BMI. Indeed, there are clear differences between MS and controls across body compartments based on DEXA, such that people with MS have higher levels of adiposity, lower BMD, and less lean soft tissue mass [58]. Such insights contrast with prior research involving BMI, where BMI was significantly lower in people with MS than non-MS controls [57]. These results challenge the value of BMI among people with MS and may accordingly underestimate adiposity and overestimate bone and/or muscle mass. This is crucial regarding the current assessment and classification of obesity, comorbidity risk, and applicable disease management in MS. Researchers may seek to establish body composition cut-points that fit the context of MS to guide clinical assessment and optimize disease-related outcomes. Additionally, there is little evidence regarding underlying factors that influence body composition in MS, and this should be an avenue for future research. Such research might be guided by the social ecological model, and examine multilevel factors including individual, social, and societal/environmental that contribute to worse body composition in MS [125]. Such studies may assist in reframing how health in MS is assessed and managed from an individual and public policy perspective to promote evidence-based personalized care.

The current evidence regarding body composition and putative outcomes using DEXA appears nuanced and sometimes inconsistent in MS, although some trends are notable (Figure 1 and Appendix A). There is some evidence that greater adiposity may be related to higher levels of CRP, poorer mobility outcomes, symptoms such as greater depression and pain, as well as worse cardiorespiratory fitness. Higher bone mass (i.e., higher BMD and/or BMC) appears to have some relationship with better cognitive outcomes, greater muscular strength, and higher physical and psychological HRQOL. Higher lean soft tissue mass is seemingly related to better cognitive and mobility outcomes and greater muscular strength.

These trends should be approached with caution, as some categories of outcomes were better represented in the literature than others. This limitation should be addressed in future research, such that studies may examine the link between body composition and blood-based biomarkers of inflammation, neuroaxonal damage, and brain structure and function based on magnetic resonance imaging. Importantly, the majority of results presented are associative in nature and may not necessarily be interpreted such that body composition is the relative cause of these outcomes. Further, many studies included only participants that were either recently diagnosed or had relapsing–remitting MS, and future studies should seek to be as inclusive as possible prior to generating conclusions regarding body composition and outcomes in MS. Another consideration is that body composition may be a product of the MS disease course and/or its consequences. Future research should investigate these outcomes as an approach for appropriately guiding research on the management of outcomes in MS.

The preliminary evidence regarding interventions involving physical activity and/or diet on body composition outcomes is quite promising, particularly given that body composition was not always the primary outcome in the studies presented in this narrative review. The results across studies should be replicated in future research, with a particular emphasis on tracking the long-term effects and feasibility of maintaining changes in body composition with health-promoting behaviors across the range of disease courses [123]. This body of research should further characterize changes in secondary outcomes associated with the physical activity and dietary interventions that successfully change body composition. Future research might also include more body composition components and compartments, as only one physical activity study examined the impact on bone and the vast majority reported only whole-body outcomes across tissue components [110].

## 7. Conclusions

This paper provided a focus on body composition followed by a narrative review of published research on body composition and its measurement, outcomes, and interventions in MS. The narrative review provided an overview and organization of the research and a road map that guides directions for future research based on gaps in knowledge. The narrative review provides a clear rationale for including comprehensive measures of body composition in MS. This narrative review of the current knowledge base indicates that researchers should prioritize continued and focal research on body composition in MS, as this represents a potent and fruitful line of research for informing the clinical care of MS.

## Figures and Tables

**Figure 1 nutrients-17-01021-f001:**
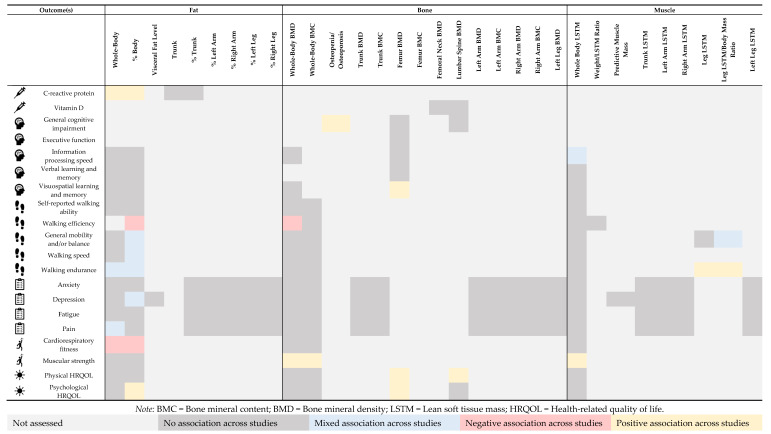
Body composition via DEXA and disease- and health-related outcomes in multiple sclerosis (evidence map).

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
