# Peer review of "Body Composition and Its Outcomes and Management in Multiple Sclerosis: Narrative Review"

_nutrients, 2025, doi:10.3390/nu17061021_

Round 1
Reviewer 1 Report
Comments and Suggestions for Authors
Review: Body Composition and Its Outcomes and Management in 2 Multiple Sclerosis
This article is a review of obesity markers in multiple sclerosis. It is an interesting article, but it needs some major revisions. The article in general does not follow a typical structure for review studies. There are no well-defined sections, and it does not provide methodological information and there is no criterion for reviewing only two meta-analyses.
Abstract
The abstract does not follow a typical structure of research articles, in this case a review. Please add the objective of the review, indicate the type of review that has been carried out, where the search was made, and the main results.
Introduction
Lines 67-81. This paragraph seems like the methodology of the study. This last paragraph should justify the study. It should answer: What does this review contribute to the previous literature on the topic? State the research question and its objectives.
General comments:
- There is no methodology section: This should include the design (type of review conducted), search strategy (if it is not a narrative review), when it was conducted and how. Are there any quality criteria?
- Why are only two meta-analyses reviewed?
- Points 2,3,4,5 should go within a results section.
- The sections seem to be a description of the results explored but do not respond to a direct relationship with the outcome of interest (multiple sclerosis).
- The limitations and strengths of the study are also not indicated.
- The conclusion does not respond to the "objective of the review", and a conclusion cannot be so extensive.
Authors should review the structure of this type of articles and try to adapt and synthesize it.
Comments on the Quality of English LanguageI am not qualified to assess English.
Reviewer 2 Report
Comments and Suggestions for Authors
The article explores the role of body composition in managing and understanding multiple sclerosis (MS). It critiques the reliance on BMI for assessing obesity and highlights dual-energy X-ray absorptiometry (DEXA) as a superior method for measuring body composition. The review includes meta-analyses comparing body composition between MS patients and non-MS controls, showing differences in fat, bone, and muscle mass. It discusses how body composition affects MS-related outcomes like cognition, mobility, symptoms, and quality of life and evaluates the impact of physical activity and diet-based interventions on body composition.
1. Provide definitions or additional context for technical terms like "DEXA," "BMI," and "body composition models" early in the article for readers unfamiliar with these concepts. Include more details on how studies were selected for the review and meta-analyses to strengthen transparency.
2. Simplify tables and charts or provide summaries for complex datasets to improve accessibility for general readers. Ensure figures and tables are easy to interpret and include concise legends or summaries to help readers quickly grasp key points.
3. Reinforce how the review advances understanding of MS management and suggest specific areas for future research. Expand the introduction to provide a broader context of obesity, body composition, and MS to set the stage for the review.
4. Provide actionable recommendations for clinicians and researchers based on the findings, such as specific body composition targets or preferred intervention strategies.
Round 2
Reviewer 1 Report
Comments and Suggestions for Authors
Review: Scoping Review of Body Composition and Its Outcomes and Management in Multiple Sclerosis
General comment: The authors report that it is a review work, but it does not follow guidelines and does not meet any quality criteria of scoping reviews.
Abstract:
In methods: What inclusion criteria were followed? Type of studies included?
In results: How many items were identified? How many were reviewed.
Introduction:
Modify the text: The last paragraph of the introduction is not appropriate. In a review work, especially in a scoping review, you must indicate what this review contributes to the previous literature, what your research question is and finally end with your objective.
Methods:
The methodology of a scoping review must be concise, clear and reproducible.
You must add sections:
-Design, when the search was carried out and what guidelines it follows (Prisma, Cochrane?).
-search strategy: how it was searched and how many results were obtained. How many researchers carried out the search.
-Review Criteria and Study Selection
-Data Extraction
-Quality Assessment (not mandatory).
Results:
-There is no flowchart
- Tables according study characteristics?
Limitations:
The proposed limitations are from their study, not from the review work itself.
Reviewer 2 Report
Comments and Suggestions for Authors
The author respond to the comments clear and the manuscript is well written. Thank you for the opportunity of review this work.
